# Preoperative Crohn’s Disease Exclusion Diet and Exclusive Enteral Nutrition in Adults with Crohn’s Disease: A Feasibility Randomised Controlled Trial

**DOI:** 10.3390/nu16132105

**Published:** 2024-07-02

**Authors:** Catherine L. Wall, Rachael Bensley, Tamara Glyn, Melissa Haines, David Rowbotham, Ian Bissett, Timothy Eglinton, Richard B. Gearry

**Affiliations:** 1Department of Medicine, University of Otago, Christchurch 8011, New Zealand; 2Department of Surgery and Critical Care, University of Otago, Christchurch 8011, New Zealand; 3Christchurch Hospital, Te Whatu Ora Health New Zealand Waitaha Canterbury, Christchurch 8011, New Zealand; 4Waikato Hospital, Te Whatu Ora Health New Zealand Waikato, Hamilton 3240, New Zealand; 5Auckland City Hospital, Te Whatu Ora Health New Zealand Te Toka Tumai, Auckland 1023, New Zealand; 6Department of Surgery, The University of Auckland, Auckland 1023, New Zealand

**Keywords:** Crohn’s disease, surgery, nutritional optimization, exclusive enteral nutrition, Crohn’s disease exclusion diet, oral nutrition supplements, surgical complications, preoperative rehabilitation

## Abstract

Preoperative exclusive enteral nutrition (EEN) improves nutritional status, reduces intestinal inflammation, and likely improves surgical outcomes. Crohn’s disease exclusion diet with partial enteral nutrition (CDED) also reduces intestinal inflammation but its safety preoperatively is unknown. This single-blinded, multicentre, randomised controlled trial of three preoperative nutritional therapies aimed to assess the feasibility of recruiting and retaining patients and collecting primary and secondary effectiveness outcomes. Adults undergoing elective Crohn’s disease surgery with a body mass index (BMI) > 18.5 kg/m^2^ and without significant weight loss were eligible to participate. Patients were randomly assigned to six weeks of preoperative EEN, CDED, or standard care. Feasibility, nutritional, radiological, and surgical outcomes were recorded. Over 18 months, 48 patients were screened, 17 (35%) were randomised, and 13/17 (76%) patients were retained in the intervention phase. It was feasible to collect primary and secondary effectiveness data; at day 30, three patients had Clavien Dindo Grade 2 complications, and 10 had no complications. Nutritional therapy adherence of patients retained in the study was high. Recruitment and retention of patients who need elective Crohn’s disease surgery for preoperative nutritional therapy is possible, although a shorter duration may improve EEN completion. The impact on surgical outcomes should be assessed in a larger study.

## 1. Introduction

Half of patients with Crohn’s disease will require gastrointestinal surgery during their lifetime. Elective, rather than emergency, surgical procedures allow for optimisation of medications, inflammation, and nutritional status prior to surgery [1]. Nutrient deficiencies and impaired nutritional status are common in adults with Crohn’s disease [2,3]. Patients awaiting surgery are at greater risk of poor nutritional status due to active disease, strictures, and/or penetrating disease, all of which affect patients’ ability to eat normally. Malnutrition, defined as either body mass index (BMI) < 18.5, >10% body weight loss in the last 3–6 months, or low muscle mass, is an independent risk factor of poor surgical outcomes, such as death, surgical complications, and postoperative infections [4,5]. Current guidelines recommend that all patients undergoing gastrointestinal surgery are screened for malnutrition and that those who are malnourished or at risk of malnutrition are nutritionally optimised preoperatively with 600 kcal/day of partial enteral nutrition to improve surgical outcomes [6]. However, in patients without malnutrition, it is not known whether preoperative nutrition optimisation confers surgical outcome benefits.

Two nutritional treatments that are currently, or increasingly, used to treat Crohn’s disease inflammation in adults are exclusive enteral nutrition (EEN) and phase 1 of the Crohn’s disease exclusion diet with 50% of energy from partial enteral nutrition (CDED). EEN is a well-established treatment [7] that involves exclusive consumption of a medical oral nutritional supplement drink and avoidance of usual foods and fluids except water. Several retrospective case series suggest that preoperative EEN may improve nutritional status, reduce inflammatory burden and disease severity, and, consequently, improve surgical outcomes in patients with Crohn’s disease [8,9,10,11,12,13,14]. CDED has been proposed as an efficacious and well-tolerated alternative to EEN to treat active Crohn’s disease in adults and children [15,16], but its use in patients with stricturing or penetrating Crohn’s disease and/or prior to surgery has not previously been explored.

Currently, there are no prospective randomised controlled trials (RCTs) investigating the use of CDED or EEN in patients requiring elective surgery for the management of Crohn’s disease. The routine use of preoperative nutritional therapies has the potential to provide significant healthcare savings and quality of life improvements for patients. High-quality scientific evidence from RCTs is required to change clinical guidelines and, thus, lead to any changes in routine clinical practice. The aim of this research was to determine whether it is feasible to, firstly, recruit and retain patients with Crohn’s disease undergoing colorectal surgery in a randomised single-blinded study of nutritional therapies and, secondly, collect the planned primary and secondary effectiveness outcomes.

## 2. Materials and Methods

Trial design. This three-arm feasibility RCT was conducted at three hospitals (Christchurch, Auckland City, and Waikato) in New Zealand. It was registered (ACTRN12621000002886), received ethical approval (Northern B Health and Disability Ethics Committee, New Zealand, 20/NTB/134), and locality authorisations were obtained. Participants provided informed consent prior to participation.

The initial recruitment sites were Christchurch and Auckland City Hospitals, but prolonged COVID-19 lockdowns in Auckland resulted in cancellations of elective surgeries. An additional site (Waikato Hospital) was included to increase patient recruitment (Ethics amendment reference 20/NTB/134/AM01).

Participants. Potentially eligible patients were identified by colorectal surgeons and gastroenterologists at each hospital. Patients aged >18 years who required elective gastrointestinal surgery to manage Crohn’s disease were eligible to participate. Patients were excluded for any of the following reasons: (1) body mass index <18.5 kg/m^2^, (2) weight loss >10% body weight in the last six months, (3) the need for pre-surgical parenteral nutrition, (4) self-induced dietary restriction preventing adherence to the treatment arms, (5) surgery planned <6 weeks, (6) did not speak English, and (7) had a stoma. During the first six months of recruitment, the exclusion criteria were modified to include patients with an existing stoma who needed elective surgery to manage Crohn’s disease upstream of the stoma (Ethics amendment reference 20/NTB/134/AM04).

Interventions. Patients were randomised to one of three treatment arms (allocation ratio 1:1:1) for six weeks prior to surgery. All study patients, regardless of intervention allocation, received four telephone follow-ups (weeks 1, 2, 4, and 5) with a research dietitian to assess tolerance and adherence to the interventions, severity of gastrointestinal symptoms, and adequacy of nutritional intake. Medical therapy was managed by the treating gastroenterologist in consultation with the colorectal surgeon. Corticosteroids were weaned to the minimum dose possible, while biologic medications were withheld for one dose interval prior to surgery. Immunomodulators were continued through the perioperative period.

Exclusive enteral nutrition. Oral EEN provided 100% of nutritional requirements from polymeric oral nutritional supplements (ONS) (e.g., Abbott Nutrition (Hoofddorp, The Netherlands) Ensure Plus^®^ or Nutricia (Hoofddorp, The Netherlands) Fortisip^®^). Patients were instructed to avoid consumption of usual foods and fluids, with the exception of water and black tea or coffee. All patients randomised to EEN received an initial education via video consultation and four telephone follow-ups by one independent, EEN-experienced gastroenterology dietitian. Nutritional requirements were estimated at 30–35 kcal/kg/day and adjusted throughout the intervention period according to appetite and weight. The ONS were provided on prescription and prescription costs were paid by the research funding.

Phase 1 Crohn’s disease exclusion diet. The CDED is a multi-phased treatment. Phase 1 is a six-week induction of remission phase comprising 50% of energy intake from oral partial enteral nutrition and 50% of energy from specific foods [15]. Phase 2 comprises 25% of energy intake from oral partial enteral nutrition and 75% of energy from an increased range of specific foods [15]. Phase 1 CDED was chosen as it has been shown to treat active Crohn’s disease inflammation [15]. Patients randomised to CDED received verbal education and written resources from a research dietitian at each site. Patients also had access to Nestlē Health Science Modulife© website/App recipes and resources. Standard ONS was prescribed to meet half the nutritional requirements (15–20 kcal/kg/day), and the remaining 50% of nutritional requirements were to be met by the CDED mandatory and allowed foods and fluids. Patients were instructed to avoid non-allowed foods and fluids. Patients received telephone follow-ups by a research dietitian four times during the six-week treatment.

Standard Care. In New Zealand, patients are not routinely provided with dietary advice preoperatively but, anecdotally, ask what they should eat while waiting for elective surgery. For a multicentre trial, it was necessary to define and standardise standard care. Standard care was defined as a soft, textured, high-energy, high-protein diet as this would be appropriate for most surgical indications in patients with Crohn’s disease. Standard care was not defined as a low-fibre or low-residue diet as usual dietary fibre intake is often already lower than recommended [17]. Patients were provided with written resources and verbal education from the site research dietitian. The research dietitian assessed nutritional intake at baseline and during the six-week intervention during four telephone follow-up calls. If patients could not meet their nutritional requirements from food alone, they could drink up to 600 kcal per day (20–30% of their nutritional requirements) from a standard ONS. This volume of ONS can help maintain nutritional status but is unlikely to reduce intestinal inflammation [18].

Outcomes. Outcome measures were assessed at baseline prior to nutritional interventions (week 0); pre-surgery (week 6); peri-surgery (week 7–8); 30 days post-surgery (week 11–12); and three months post-surgery (week 18). Standardised and validated tools, where available, were used.

Primary outcomes. The main outcomes were the feasibility and acceptability of the preoperative nutritional intervention. Feasibility was assessed by (i) the number of patients recruited, (ii) retention of patients, (iii) verbal self-reported intervention adherence during telephone follow-ups, (iv) tolerance of interventions assessed by research dietitians as the patient’s ability to adhere to a prescribed intervention without exacerbation of gastrointestinal symptoms or inadequate nutrition intake that results in significant weight loss, and (v) surgeon report (a two-question electronic survey) of blinding success. Acceptability of the preoperative nutritional interventions was assessed in one of two patient video focus groups (Zoom Video Communications Inc. (San Jose, CA, USA), version 5.13.11), which were audio recorded and transcribed. A focus group guide included open-ended, semi-structured questions. Participants unable to attend completed the questions by email.

Effectiveness outcomes. The primary effectiveness outcome was a 30-day post-surgical complication rate assessed and classified according to the Clavien Dindo classification [19]. Data on immediate postoperative complications were collected from the hospital discharge summary. Complications post-hospital discharge were assessed during a telephone call on day 30 post-surgery.

Secondary effectiveness outcomes were: (i) intraoperative stoma formation, (ii) hospital length of stay, and (iii) change in intestinal mucosal damage (assessed by magnetic resonance imaging (MRI) within three months of the baseline assessment and a week prior to surgery. Damage was scored by the radiologists according to the sMaRIA score [20], (iv) change in clinical disease activity (Harvey Bradshaw index) [21], and biomarkers of intestinal inflammation (faecal calprotectin and *C*-reactive protein), and (v) quality of life (UK-IBDQ) [22]. Serious adverse reactions in response to the nutritional therapies were not anticipated.

Other outcomes. At baseline, week 6 and week 18, patients had a nutritional assessment conducted by the research dietitians. The assessment included anthropometry (weight and height), biochemistry, and dietary intake. Participants with blood micronutrient results below the normal range were considered deficient and prescribed a micronutrient supplement to correct deficiency according to usual standard care. Dietary intake was assessed using an estimated 4-day food and beverage record. Food-related quality of life was assessed using the validated FR-QoL-26 questionnaire [23,24].

Sample size. A formal sample size calculation was not completed as the purpose of this study was to assess the feasibility of the nutritional interventions and study design. A target sample size of 20–25 patients was informed by the expected number of eligible elective surgeries in an 18-month recruitment period (10–15 per site) and allowed for a 0.8 (95% CI (0.639, 0.915)) retainment rate.

Randomisation and Blinding. An independent statistician not involved in the trial created a block randomisation file. Blocks of six (2 × 3 interventions), stratified per site, were created in a Microsoft Excel for Mac (Version 16.96)^®^ file that was uploaded into a custom-built, secure, password-protected Research Electronic Data Capture (REDCap) database (v12.2.6. 2023 Vanderbilt University). The research dietitians and trial coordinator were blinded to the intervention sequence. At baseline, patients were consecutively randomised to an intervention group by the site research dietitian in REDCap. From this point the participant and research dietitian providing the nutritional therapy education were aware of the treatment allocation. In an attempt to blind surgeons to the preoperative nutritional therapy allocation, participants were instructed not to discuss their treatment with their colorectal surgeon or treating gastroenterologist.

Analytical methods. Study data were stored in REDCap databases. The feasibility study was not designed or powered to find a difference between the groups. Data were exported from REDCap into Microsoft Excel^®^. Quantitative data were summarised as counts, medians, and interquartile ranges. Dietary intake data from the estimated 4-day food and beverage dairies were entered by the research dietitians into the nutrient analysis programme Foodworks (Xyris Software (Version 10) Australian/NZ Foodfiles). Non-parametric t-tests were used to test for between-group differences in preoperative dietary intake. Qualitative data from focus group transcriptions were analysed thematically, and selected representative quotes were presented.

## 3. Results

The baseline characteristics of participants enrolled in this study are presented in Table 1. Participants had a median age of 37.9 years (IQR 35.9 to 57.1 years) and there was a similar proportion of males and females recruited. Most (65%) participants had stricturing disease.

### 3.1. Feasibility Outcomes

#### Participant Recruitment and Retention

Over the 18-month recruitment period (September 2019 to March 2022), 48 patients were screened, 24 (50%) were ineligible, seven (15%) were eligible but declined to participate, and 17 patients (35%) were eligible and randomised (Figure 1). Four (24%) patients withdrew during the six-week intervention period (retainment of 0.76); one in the usual care group had a preoperative myocardial infarction at week 4 and three in the intervention groups at weeks 1, 2, and 5 due to poor mental health (Figure 1). The three-month post-surgery research visit was attended by 11/17 (65%) participants randomised at baseline.

### 3.2. Intervention Adherence and Tolerance

The study groups had similar baseline characteristics (Table 1), except that there was a higher proportion of females in the EEN group (five out of six patients). Usual care was well tolerated without exacerbation of gastrointestinal symptoms, but two participants reported a poor appetite and consumed up to 600 kcal/day from ONS to prevent weight loss. EEN treatment was less well tolerated; four out of six patients withdrew or changed treatment (Figure 1). One participant withdrew in week 1 due to low mood and energy. One patient continued to eat usual foods and withdrew at week 5 due to deterioration in mental health. Two participants stopped EEN within the first two weeks and changed to CDED or usual care groups (Figure 1). CDED was well tolerated without exacerbation of gastrointestinal symptoms or significant weight change, and four out of five (80%) participants reported high adherence (>90% of total energy intake from CDED). One patient, who had tolerated the CDED intervention well, withdrew from the study the week of surgery due to increasing surgery-related anxiety. All participants, regardless of preoperative intervention allocation, commented positively about the dietitian input and support they received. “I felt totally supported and could reach out if and when I needed”. Those on nutritional therapy (EEN, CDED) commented on the improvement of their Crohn’s symptoms while following the diet. “That diet changed my toilet motions dramatically and I felt good”.

Participants in the EEN and CDED groups consumed more calories, protein and carbohydrates and less saturated fat compared with the usual care group (Table 2). Total dietary fibre intake was higher in the CDED group compared with the usual care and EEN groups and did not result in increased gut symptoms.

### 3.3. Surgeon Blinding

Colorectal surgeon blinding was possible. Patients were operated on by six different surgeons. Surgeons were asked if they were informed which treatment patients received preoperatively and also which treatment they thought patients received. Responses were received for 14/17 patients; surgeons commented that in most cases, they did not enquire which treatment patients had been allocated, and neither did patients inform them. Surgeons also could not correctly identify which treatment patients were allocated to, except for the two patients on EEN who withdrew from the study.

### 3.4. Collection of Efficacy Outcomes

In all cases, it was possible to collect the primary efficacy outcome of Clavien Dindo classification at day 30 and other surgical outcomes (Table 3). It was more challenging to collect objective markers of inflammation (Figure 2); either participants did not bring stool samples to the study appointment, or COVID-19 restrictions meant that in-person appointments and sample collection were not possible. It was possible to collect electronic questionnaire data (via an email link or at appointments) and usual care bloods as this service was less impacted by COVID-19 research restrictions (Figure 2).

There was variability in baseline inflammation markers (CRP, faecal calprotectin), disease symptoms (HBI), and quality of life scores (Figure 2). All data points are presented together, given the small sample size of the intervention groups. There were no consistent changes in these markers associated with the six-week intervention.

## 4. Discussion

This is the first RCT to use CDED in a preoperative setting. It was feasible for participants to follow CDED, which included a moderate amount of fibre (Table 3) from whole, peeled fruit and vegetables, including raw salad vegetables. As previously shown [18], randomisation to EEN may be associated with lower study retention. The findings of this trial support the development of a powered RCT to investigate effective preoperative nutritional optimisation regimens to reduce the risk of postoperative complications.

Two-thirds of potentially eligible patients were ineligible for the trial, most commonly due to the inability to identify potentially eligible patients at least six weeks prior to elective surgery. Reasons for this included a lack of IBD MDT meetings (i.e., surgeons, gastroenterologists, radiologists, pathologists, clinical dietitians, IBD nurses, and clinical researchers) [26], prioritised discussion of urgent surgeries (e.g., bowel cancer surgery over elective Crohn’s disease surgery) at colorectal/radiology meetings, variable surgical waitlists, and a significant cyber crisis which limited access to electronic medical records.

The total number of elective Crohn’s disease surgeries performed during the recruitment period was negatively impacted by the COVID-19 coronavirus outbreak. Multiple COVID lockdowns in Auckland meant there was less capacity to conduct elective surgery. In Waikato, surgical theatre space was already limited due to COVID-19 lockdown surgery cancellations, and the cyber crisis further delayed elective surgery wait times, as cancer surgery took precedence [27]. Furthermore, in September 2021, bowel cancer screening was rolled out in Canterbury, New Zealand, increasing the number of urgent cancer surgeries, which resulted in fewer theatre spots for elective Crohn’s disease surgeries and significantly impacted study recruitment.

The retention of patients in the study was as expected during the intervention phase; three-quarters remained in the study, although missing data at the three-month post-surgery follow-up time point did occur. The main reasons for withdrawal were related to poor mental health. Poor mental health is common in active IBD [28], and in a larger trial, screening for anxiety or depression and exclusion of patients with severe symptoms may improve study retention.

The feasibility study sample size was inadequate to provide sufficient data to calculate a sample size for a future RCT. It appears that partial enteral nutrition with phase 1 CDED is safe preoperatively and could be included in a powered RCT. Previous data [18,29] and our experience in this study suggest that adherence to nutritional therapy is patient-dependent and may be influenced by age and sex. It was less feasible to randomise patients to six weeks of EEN. The optimal duration of preoperative EEN is unknown. Six weeks was chosen as this is the minimum recommended induction of remission duration [30]. In a non-surgical adult cohort, two weeks of EEN (*n* = 32) significantly improved markers of inflammation (CRP, faecal calprotectin, and Harvey Bradshaw index) [18]. A longer course of mostly enteral nutrition may also confer surgical outcome benefits. A mixed cohort of malnourished and well-nourished patients who consumed at least two weeks of enteral nutrition (defined as ≥600 kcal/day from ONS) had significantly fewer surgical complications (37/204 [18.1%] vs. 36/96 [37.5%], *p* < 0.001) compared with a non-optimised group [31]. Interestingly, there was no difference (*p* = 0.99) in the frequency of complications between patients who used EEN (*n* = 125) (defined as >75% of energy from ONS) compared with those who used partial enteral nutrition (<75% of energy from ONS) (*n* = 17) [31]. The balance between length of treatment to optimise adherence to the treatment and benefits has not yet been explored but future studies should expect withdrawal from such treatment arms in sample size power calculations and stratifying by gender and/or age may be important.

In this feasibility study, non-English speakers were excluded because education on the exclusive enteral nutrition intervention was delivered via a video call, and all participants were monitored via telephone calls. It would have been challenging to conduct this in a feasibility study setting via an interpreter. In a larger study, if the delivery of all the interventions is conducted on-site, then non-English speakers could be included with support from an interpreter and appropriate written materials.

The optimal assessment of preoperative response to nutritional therapy is not clear. There were limited complete data sets of markers of inflammation (serum CRP, faecal calprotectin, and MRE) (*n* = 11), and, in the small sample size, findings were not concordant. There are few prospective studies published. A small study (*n* = 10) with malnourished patients suggested that preoperative EEN reduces disease symptoms (HBI) and markers of inflammation (CRP, faecal calprotectin) [32]. Another study of patients at high risk for postoperative complications did not report preoperative change in disease markers [33]. Retrospective studies with larger sample sizes suggest that preoperative EEN may [9] or may not [8] be associated with a significant reduction in CRP. No studies have reported changes in faecal calprotectin or MRE imaging with the use of preoperative nutritional therapy. It is hypothesised that nutritional therapy is associated with improved surgical outcomes in part due to a reduction in inflammatory burden, which is the most sensitive marker of inflammatory burden in the preoperative setting is not yet clear.

This study was not powered to find a difference in surgical outcomes between groups. There is only one published case–control study of EEN prior to elective Crohn’s disease surgery; a retrospective case–control study comparing cases treated with EEN (*n* = 51) matched with historical controls (*n* = 78) [9]. EEN was well tolerated, with only 3/51 patients (6%) ceasing treatment, and 13 patients had clinical improvements such that they no longer needed surgery. In those optimised with at least four weeks of EEN, there were significantly fewer surgical complications: 3/38 vs. 32/78 controls (*p* < 0.001) [9]. A major limitation of retrospective studies is that patients who agree to initiate EEN are self-selected and, therefore, may be more likely to tolerate the treatment compared to patients randomised to receive EEN. A systematic review of retrospective cohort studies of preoperative EEN found that EEN was well tolerated [34]. Furthermore, retrospective studies are poorly controlled for surgery-type hospital discharge processes and prone to selection bias, as participants were not randomised to an intervention or control group.

The primary effectiveness outcome was the Clavien Dindo 30-day postoperative complications. It was feasible to collect the outcome components, and in this study, most participants had no early postoperative complications. This is not unexpected given that it was an elective surgery cohort and patients were not overtly malnourished. The role of perioperative nutritional therapy on late postoperative outcomes, including Crohn’s disease recurrence, has not been widely explored. There is evidence that gut microbiome structure may be associated with disease recurrence [35] and that low dietary fibre consumption in surgical cohorts may be associated with greater odds of disease flare [36]. Whether perioperative nutritional therapies, via postoperative gut microbiome function, can influence late postoperative complications and disease recurrence is an idea that warrants investigation.

## 5. Conclusions

It is feasible to recruit and retain patients who are not overtly malnourished and need elective Crohn’s disease surgery for a study using preoperative EEN or CDED. However, sub-groups of patients may be less likely to adhere to EEN for six weeks preoperatively. It is also feasible to collect postoperative effectiveness outcomes. The optimal duration and impact of preoperative nutritional therapy on surgical outcomes should be assessed in a larger study.

## Figures and Tables

**Figure 1 nutrients-16-02105-f001:**
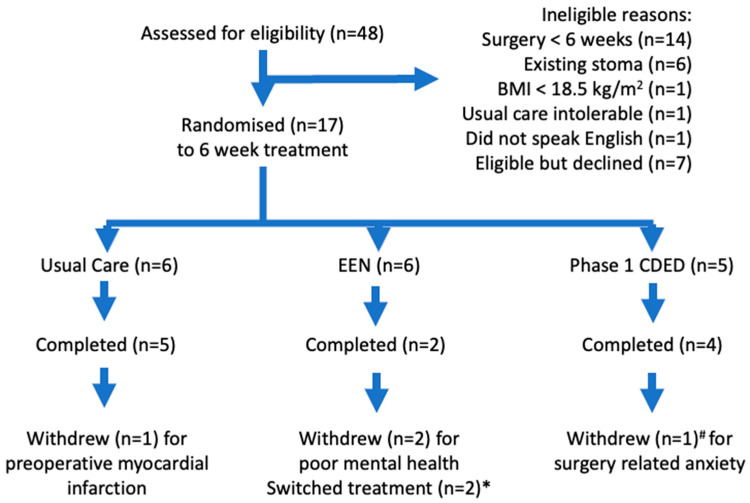
CONSORT patient flow diagram. * Switched from EEN to usual care after 5 days and to CDED after 2 weeks. # Completed at least 5 weeks of treatment prior to withdrawing.

**Figure 2 nutrients-16-02105-f002:**
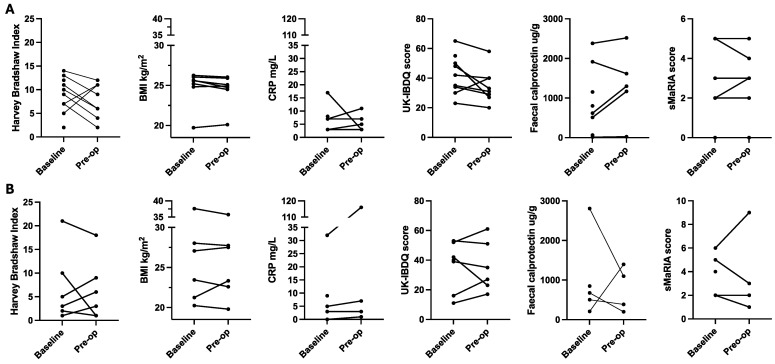
Preoperative change in markers of inflammation, Crohn’s disease symptoms, quality of life, and body mass index ((**A**) nutritional therapy; (**B**) standard care). Note: Figures include all available data points including baseline data of patients who did not complete the study. BMI: body mass index; CRP: c-reactive protein; sMaRIA: simplified magnetic resonance index of activity; UK-IBDQ: United Kingdom inflammatory bowel disease quality of life.

**Table 1 nutrients-16-02105-t001:** Baseline characteristics of study participants.

		Intervention Group		
Characteristics	All Participants (*n* = 17)	EEN (*n* = 6)	CDED (*n* = 5)	Usual Care (*n* = 6)
Age (years)—Median (IQR)	37.9 (35.9, 57.1)			
Median (range)		33.9 (27.0, 60.8)	37.3 (28.4, 68.2)	42.2(35.9, 67.8)
Female *n* (%)	9 (53)	5 (83)	2 (40)	2 (33)
BMI kg/m^2^—Median (IQR)	25.7 (24.5, 27.3) #			
Median (range)		25.6 (19.7, 28.7)	25.6 (24.8, 35.8)	26.4 (20.3, 37.6)
Crohn’s disease phenotype *n* (%)				
- Inflammatory (B1)	1 (6)	0 (0)	1 (20)	0 (0)
- Stricturing (B2)	11 (65)	2 (33)	3 (60)	6 (100)
- Penetrating (B3)	5 (29)	4 (66)	1 (20)	0 (0)
- Perianal	3 (18)	2 (33)	0 (0)	1 (20)
Previous surgery *n* (%)	5 (29)	3 (50)	1 (20)	1 (17)
Crohn’s disease medication ^ *n* (%)				
- Immunosuppressant	10 (59)	2 (33)	4 (80)	4 (67)
- Anti-TNF	9 (53)	4 (67)	3 (60)	2 (33)
- No IBD medication	2 (12)	1 (17)	0 (0)	1 (17)
Smokes/Vapes (Yes) *n* (%)	4 (24)	1 (17)	1 (20)	2 (33)
Disease activity (HBI) *n* (%)				
Median (IQR)	7 (5, 11)			
- >5	11 (65)	3 (50)	5 (100)	3 (50)
Faecal calprotectin (μg/g) *n* (%)				
Median (IQR)	677 (504, 1153) #			
- <500	3 (17)	0 (0)	2 (40)	1 (17)
- 500–1000	6 (35)	2 (33)	1 (20)	3 (50)
- >1000	4 (24)	2 (33)	1 (20)	1 (17)
- No stool sample	4 (24)	2 (33)	1 (20)	1 (17)
Raised CRP (>3 mg/L) *n* (%)	8 (47)	2 (33)	3 (40)	3 (50)
Median (IQR)	3 (3, 7)			
Micronutrient deficiency *n* (%)				
- Vitamin D	5 (33) #	1 (20) *	2 (40)	2 (40) *
- Folate	2 (13)	1 (17)	0 (0)	1 (20) *
- Vitamin B12	4 (24)	3 (50)	1 (20)	0 (0)
- Iron	5 (29)	2 (33)	1 (20)	2 (33)
UK-IBDQ score				
Median (IQR)	40 (30, 50)	36 (25, 48)	35 (34, 48)	41 (39, 50)
FR-QoL-28 score Median (IQR)	69 (62, 75)			
- Impaired: score < 90 *n* (%)	15 (88)	5 (83)	4 (80)	6 (100)

BMI: body mass index; CRP: c-reactive protein; FR-QoL-28: food-related quality of life questionnaire; HBI: Harvey Bradshaw index; IQR: interquartile range; UK-IBDQ: United Kingdom inflammatory bowel disease quality of life questionnaire. ^ No patients were taking corticosteroids. # Less than 17 data points available. * Only 5/6 patients had baseline blood taken. Micronutrient deficiency definitions; vitamin D < 50 nmol/L; folate < 8 nmol/L; vitamin B12 (cobalamin) < 170 pmol/L; iron (ferritin) < 30 μg/L if CRP < 10 mg/L or <100 μg/L if CRP > 10 mg/L [25].

**Table 2 nutrients-16-02105-t002:** Surgical procedure and surgical outcomes.

Efficacy Outcomes ^1^	All Participants (*n* = 13)	EEN (*n* = 4) *	CDED (*n* = 4)	Usual Care (*n* = 5)
Surgical procedure, *n* (%)				
- Ileal resection	1	0	0	1 #
- Ileocolic resection	10	3	4	3 #
- Ileosigmoid resection	1	1	0	0
- Proctocolectomy	1	0	0	1
- Subtotal colectomy	1	0	0	1
- Stricturoplasy	1	0	0	1 #
Complications, *n* (%)				
- None	10 (77)	4 (100)	3 (75)	3 (60)
- Grade I	0 (0)	0 (0)	0 (0)	0 (0)
- Grade II	3 (23)	0 (0)	1 ^a^ (25)	2 ^b,c^ (40)
Hospital length of stay (days), median (IQR)	5 (5, 6)			
median (range)		5 (4, 6)	5 (5, 6)	7 (5, 15)
Intra-operative stoma, *n* (%)				
- Planned	2 (15)	1 (25)	0 (0)	1 (20)
- Unplanned	0 (0)	0 (0)	0 (0)	0 (0)
Wound infection, *n* (%)	1 (8)	0 (0)	1 (25)	0 (0)
Other infectious complications, *n* (%)	0 (0)	0 (0)	0 (0)	0 (0)

^1^ Excludes the four patients who withdrew from the study prior to surgery. * Includes patients who changed treatment groups. # One patient had three procedures during one surgical episode. Complication details: ^a^ blood transfusion postoperatively, wound infection requiring antibiotics, ^b^ fever, clostridium difficile infection requiring antibiotics, ^c^ postoperative ileus requiring total parenteral nutrition CDED, phase 1 Crohn’s disease exclusion diet with partial enteral nutrition; EEN: exclusive enteral nutrition.

**Table 3 nutrients-16-02105-t003:** Preoperative (week 6) nutrition intake, food-related quality of life, and body mass index.

Nutrient	EEN (*n* = 2) Mean (SD)	CDED (*n* = 5) Mean (SD)	Usual Care (*n* = 5) ^1,2^ Mean (SD)	*p*-Value ^3^ (CDED vs. Usual Care)
Energy (kcal)	2555.1 (204.9)	2469.1 (761.2)	1720.0 (181.5)	0.016
- kcal/kg/day	41.5 (8.1)	30.9 (8.8)	21.4 (5.9)	0.095
Protein (g)	106.6 (8.3)	134.5 (34.4)	83.2 (14.8)	0.008
- Protein g/kg/day	1.7 (0.3)	1.7 (0.45)	1.0 (0.2)	0.008
Fat (g)	83.7 (6.9)	78.8 (31.0)	71.0 (7.5)	0.738
- Saturated fat (g)	7.7 (0.5)	11.3 (4.3)	24.7 (8.3)	0.016
Carbohydrate (g)	343.4 (28.6)	293.2 (96.0)	170.5 (35.2)	0.032
- Sugars (g)	112.9 (5.8)	110.6 (26.4)	65.0 (24.7)	0.056
Dietary fibre (g)	0 (0)	22.0 (10.8)	8.5 (2.6)	0.008
FR-QoL-26 score	67.0 (14.6)	88.0 (15.5) *	69.7 (25.9)	0.171
BMI (kg/m^2^)	22.4 (3.3)	25.3 (0.7)	26.1 (5.6)	1.000
BMI change (kg/m^2^)	−0.25 (0.9)	−0.29 (0.3)	−0.15 (1.3)	0.404

^1^ One participant did not return a food diary at the preoperative appointment. ^2^ Two participants drank 600 kcal/day from oral nutrition supplements in addition to usual foods and fluids. ^3^ Unpaired Mann–Whitney *t*-test, statistical significance assumed *p* < 0.05. * One participant did not complete the questionnaire. CDED: phase 1 Crohn’s disease exclusion diet with partial enteral nutrition; EEN: exclusive enteral nutrition; FR-QoL-26: food-related quality of life questionnaire; SD: standard deviation.

## Data Availability

The data presented in this study are available on request from the corresponding author. The data are not publicly available due to limited data use consent by study participants.

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
