# Peer review of "Preoperative Crohn’s Disease Exclusion Diet and Exclusive Enteral Nutrition in Adults with Crohn’s Disease: A Feasibility Randomised Controlled Trial"

_nutrients, 2024, doi:10.3390/nu16132105_

Round 1

Reviewer 1 Report

Comments and Suggestions for Authors

-            Use oxford comma

-            “EEN is a well-established treatment[7] that involves exclusive consumption of a medical 50 oral nutritional supplement drink”

Using nasogastric tube? How many adults are able to cover all the necessary calories with only oral nutritional supplement drink per mouth?

-            Report the specific short and middle term surgical intervention’s complications in the three groups

-            In figure 2 divide the three groups

Comments on the Quality of English Language

Native English speakers

Author Response

Use oxford comma

 We have tried to add coma as suggested.

-            “EEN is a well-established treatment[7] that involves exclusive consumption of a medical 50 oral nutritional supplement drink”

 We are not sure what change the reviewer is suggesting for this sentence.

Using nasogastric tube? How many adults are able to cover all the necessary calories with only oral nutritional supplement drink per mouth?

 No patients used a nasogastric tube. In lines 95 and 96 of the methods, we state “oral EEN provided 100% of nutritional requirements from polymeric oral nutritional supplement”. We have added “oral” to lines 104 and 106 that describes the use of partial enteral nutrition with the Crohn’s disease exclusion diet.

-            Report the specific short and middle term surgical intervention’s complications in the three groups

Thank you for your comment. Table 2 reports the short and middle term surgical complications for the three groups.  We forgot to include the table footnote that describes the specific details of the complications. This has now been added.

-            In figure 2 divide the three groups. Thank you for your suggestion. We have submitted a revised Figure 2 that includes pre and post data for standard care (n=6) and nutritional therapy (n=8). We have grouped all patients who used nutritional therapy as one group because we feel that it could be misleading to report the outcomes of two patients who used EEN for 6 weeks as a separate group.

Reviewer 2 Report

Comments and Suggestions for Authors

The aims of this study are laudable since there is considerable evidence, but largely uncontrolled, that pre-operative enteral feeding reduces inflammation in Crohn’s disease, allows tailing of steroids if currently taken, and improves outcomes of surgery. It was designed as a feasibility study rather than as a conventional phase 2 study of efficacy. This may limit its interest to general readers but it will certainly be of interest to those involved in management of Crohn’s disease patients requiring surgery so I am inclined to be lenient when considering the very small number of patients who completed the intervention period (only 13). The case for publication on this basis could be better made with some expansion of the discussion to optimise future trial design as well as modification of the Abstract to make these points more clearly.

Some points need addressing:

1.     The Abstract should clearly state the primary outcome measure(s) (but see note 2 below). Currently it could be read more like the report of an underpowered trial of efficacy. 

2.     Abstract - The primary outcome measure(s) as described in the protocol registered at ANZCTR are: (i) “recruitment rate (comparing the number of potentially eligible patients to the number of patients recruited)” ; (ii) “retention rate - number of patients enrolled in each arm of the study compared to the number who complete the intervention period”; (iii) feasibility ….   “to collect the planned primary and secondary effectiveness objectives. This will be assessed by calculating the proportion of patients who completed each outcome at each of the three time points”.  These measures should be stated clearly in the abstract together with proportions achieved. These primary outcomes seem to have been expanded considerably since trial registration (Interventions p3) to include (iv) tolerance of interventions .., (v) surgeon report of blinding success – this makes a very large number of primary outcomes. Unless outcomes (iv) and (v) really have been defined in approved protocol major amendments they would more appropriately be considered as secondary outcomes.

3.     Abstract and Discussion – although I sympathise with the Abstract conclusion: “Patients who need elective Crohn’s disease surgery tolerate pre-operative EEN or CDED” I think this needs qualification as it is not convincingly supported by the data. Figure 1 shows that only 2 of 6 patients completed their 6 week period of total enteral nutrition. I think clinicians with  experience of using total enteral nutrition in adult patients with Crohn’s disease would consider 6 weeks adherence rather optimistic (it is better tolerated in children where 6 weeks is more “normal”) and it seems reasonable to conclude (as you do in Discussion)  that a future trial could shorten this period -  eg to 4 weeks at most or even 3 weeks. This point should be made in the Abstract since arguably it is probably the more important message from this study. The alternative of testing the better tolerated partial exclusion (CDED) without looking at total enteral nutrition might run a considerable risk of failing to show efficacy  - see point 4 below.

4.     Discussion - The evidence for partial enteral feeding being effective in patients with currently active Crohn’s disease is much weaker than for total enteral nutrition – more evidence supports its use as maintenance following induction of remission eg with total enteral nutrition. This could be discussed in rather more detail either in Discussion or Introduction.

5.     Table 1- definitions used for micronutrient deficiency should be stated (particularly vitamin D since there is considerable variation in this definition)

6.     Figure 2 – although numbers are very small it is frustrating not to be able to distinguish the three different treatment groups – colour coding would be helpful.

7.     Discussion – some consideration should be given to selection of primary outcome (efficacy)measure for any future trial. The Clavien Dindo 30 day post-surgical complications measure is not Crohn’s disease specific. Arguably it might have been the most approrpiate measure when patients with Crohn’s disease were more commonly coming to surgery whilst taking considerable corticosteroid doses (and I note that no patients in this study were receiving corticosteroids at time of surgery which is commendable). Some measure of Crohn’s disease recurrence eg anastomotic ulceration or stricturing assessed endoscopically at 6 months might be more revealing now though.

8.     Discussion – although it might make adherence even more difficult – a short period of enteral nutrition post-surgery might also be beneficial to allow mucosal healing – this could be considered in future trial design.

Author Response

The aims of this study are laudable since there is considerable evidence, but largely uncontrolled, that pre-operative enteral feeding reduces inflammation in Crohn’s disease, allows tailing of steroids if currently taken, and improves outcomes of surgery. It was designed as a feasibility study rather than as a conventional phase 2 study of efficacy. This may limit its interest to general readers but it will certainly be of interest to those involved in management of Crohn’s disease patients requiring surgery so I am inclined to be lenient when considering the very small number of patients who completed the intervention period (only 13). The case for publication on this basis could be better made with some expansion of the discussion to optimise future trial design as well as modification of the Abstract to make these points more clearly.

Some points need addressing:

  1. The Abstract should clearly state the primary outcome measure(s) (but see note 2 below). Currently it could be read more like the report of an underpowered trial of efficacy. 

Thank you for your assessment. We agree and have reworded the abstract accordingly.

  1. Abstract - The primary outcome measure(s) as described in the protocol registered at ANZCTR are: (i) “recruitment rate (comparing the number of potentially eligible patients to the number of patients recruited)” ; (ii) “retention rate - number of patients enrolled in each arm of the study compared to the number who complete the intervention period”; (iii) feasibility ….   “to collect the planned primary and secondary effectiveness objectives. This will be assessed by calculating the proportion of patients who completed each outcome at each of the three time points”.  These measures should be stated clearly in the abstract together with proportions achieved. These primary outcomes seem to have been expanded considerably since trial registration (Interventions p3) to include (iv) tolerance of interventions .., (v) surgeon report of blinding success – this makes a very large number of primary outcomes. Unless outcomes (iv) and (v) really have been defined in approved protocol major amendments they would more appropriately be considered as secondary outcomes.

The abstract has been updated to align with the stated outcomes.

  1. Abstract and Discussion – although I sympathise with the Abstract conclusion: “Patients who need elective Crohn’s disease surgery tolerate pre-operative EEN or CDED” I think this needs qualification as it is not convincingly supported by the data. Figure 1 shows that only 2 of 6 patients completed their 6 week period of total enteral nutrition. I think clinicians with  experience of using total enteral nutrition in adult patients with Crohn’s disease would consider 6 weeks adherence rather optimistic (it is better tolerated in children where 6 weeks is more “normal”) and it seems reasonable to conclude (as you do in Discussion)  that a future trial could shorten this period -  eg to 4 weeks at most or even 3 weeks. This point should be made in the Abstract since arguably it is probably the more important message from this study. The alternative of testing the better tolerated partial exclusion (CDED) without looking at total enteral nutrition might run a considerable risk of failing to show efficacy  - see point 4 below.

The phrase “Patients who need elective Crohn’s disease surgery tolerate pre-operative EEN or CDED” has been removed from the abstract a statement that better reflects that findings has been added “Adherence to nutritional therapy of patients retained in study was high.” We altered the conclusion and added the suggest points “Recruitment and retention of patients who need elective Crohn’s disease surgery to a preoperative nutritional therapy is possible, although a shorter duration may improve EEN completion.”

  1. Discussion - The evidence for partial enteral feeding being effective in patients with currently active Crohn’s disease is much weaker than for total enteral nutrition – more evidence supports its use as maintenance following induction of remission eg with total enteral nutrition. This could be discussed in rather more detail either in Discussion or Introduction.

The preoperative period is an opportunity to optimise nutritional status as malnutrition is an independent risk factor for poor surgical outcomes. Partial enteral nutrition is the primary method of optimising nutritional status preoperatively in non-IBD cohorts. Further details have been added to the introduction (line 65) to make this point more explicit “at risk of malnutrition are nutritionally optimised preoperatively with 600kcal/day of partial enteral nutrition to improve surgical outcomes”.

  1. Table 1- definitions used for micronutrient deficiency should be stated (particularly vitamin D since there is considerable variation in this definition)

Thank you for your comment. We have added the diagnosis criteria used as per our local lab and referenced the ECCO iron deficiency criteria.

  1. Figure 2 – although numbers are very small it is frustrating not to be able to distinguish the three different treatment groups – colour coding would be helpful.

Thank you for your suggestion. We have submitted a revised Figure 2 that includes pre and post data for standard care (n=6) and nutritional therapy (n=8). We have grouped all patients who used nutritional therapy as one group because we feel that it could be misleading to report the outcomes of two patients who used EEN for 6 weeks as a separate group.

  1. Discussion – some consideration should be given to selection of primary outcome (efficacy)measure for any future trial. The Clavien Dindo 30 day post-surgical complications measure is not Crohn’s disease specific. Arguably it might have been the most approrpiate measure when patients with Crohn’s disease were more commonly coming to surgery whilst taking considerable corticosteroid doses (and I note that no patients in this study were receiving corticosteroids at time of surgery which is commendable). Some measure of Crohn’s disease recurrence eg anastomotic ulceration or stricturing assessed endoscopically at 6 months might be more revealing now though.

Thank you for your thoughtful comment. We have added an additional paragraph to the end of the discussion. “The primary effectiveness outcome was the Clavien Dindo 30-day postoperative complications. It was feasible to collect the outcome components, and, in this study, most participants had no early postoperative complications. This is not unexpected given it was an elective surgery cohort and patients were not overtly malnourished. The role of perioperative nutritional therapy on late post-operative outcomes, including Crohn’s disease recurrence has not been widely explored. There is evidence that gut microbiome structure may be associated with disease recurrence[35] and that low dietary fibre consumption in surgical cohorts may be associated with a greater odds of disease flare.[36] Whether perioperative nutritional therapies can influence postoperative gut microbiome function and disease recurrence is an area of research.”

  1. Discussion – although it might make adherence even more difficult – a short period of enteral nutrition post-surgery might also be beneficial to allow mucosal healing – this could be considered in future trial design.

This is an interesting idea. Using EEN as an induction therapy in the non-surgical setting is not associated with longer time to flare, therefore the rationale in a postsurgical cohort is not clearly obvious.